# CERI, CEFX, and CPI: Largely Improved Positive Controls for Testing Antigen-Specific T Cell Function in PBMC Compared to CEF

**DOI:** 10.3390/cells10020248

**Published:** 2021-01-27

**Authors:** Alexander A. Lehmann, Pedro A. Reche, Ting Zhang, Maneewan Suwansaard, Paul V. Lehmann

**Affiliations:** 1Cellular Technology Ltd., Shaker Heights, OH 44122, USA; alexander.lehmann@immunospot.com (A.A.L.); ting.zhang@immunospot.com (T.Z.); maneewan.suwansaard@immunospot.com (M.S.); 2Laboratorio de Inmunomedicina & Inmunoinformatica, Departamento de Inmunologia & O2, Facultad de Medicina, Universidad Complutense de Madrid, 28040 Madrid, Spain; parecheg@med.ucm.es

**Keywords:** T cell immune monitoring, CD8+ T cell immunity, CD4+ T cell immunity, antigen presenting cell functionality, PBMC fitness, immune dominance, T cell determinant, aleatory T cell recognition, ELISPOT, FluoroSpot, ImmunoSpot, SARS-CoV-2, COVID-19

## Abstract

Monitoring antigen-specific T cell immunity relies on functional tests that require T cells and antigen presenting cells to be uncompromised. Drawing of blood, its storage and shipment from the clinical site to the test laboratory, and the subsequent isolation, cryopreservation and thawing of peripheral blood mononuclear cells (PBMCs) before the actual test is performed can introduce numerous variables that may jeopardize the results. Therefore, no T cell test is valid without assessing the functional fitness of the PBMC being utilized. This can only be accomplished through the inclusion of positive controls that actually evaluate the performance of the antigen-specific T cell and antigen presenting cell (APC) compartments. For Caucasians, CEF peptides have been commonly used to this extent. Moreover, CEF peptides only measure CD8 cell functionality. We introduce here universal CD8+ T cell positive controls without any racial bias, as well as positive controls for the CD4+ T cell and APC compartments. In summary, we offer new tools and strategies for the assessment of PBMC functional fitness required for reliable T cell immune monitoring.

## 1. Introduction

T cell monitoring relies on functional test systems. Antibodies in isolated serum are stable for years and this fact largely facilitates monitoring of humoral immunity. While long-lived in vivo [1], T cells in the blood are perishable and start dying shortly after their isolation from the body. In most cases, the blood drawn at clinical sites first needs to be transported to a test laboratory where the PBMC containing the T cells and APC essential for functional T cell assays are isolated. Excessive shear forces exerted during the drawing of the blood, a delay in its transportation, its exposure to too-cold or too-hot temperatures during transit can each cause damage to the T cells and APC, leading to an impairment of their fitness when tested [2]. As it is not practical to test the PBMC one by one as they arrive in the laboratory, most cryopreserve these cells to enable later testing in larger batches, and/or to be able to repeat test results or to extend testing as needed. During freeze-thawing, T cells and APC can also incur damage. Indeed, the development of protocols to freeze and subsequently thaw PBMCs without impairing antigen-specific CD4+ or CD8+ T cell functionality was one of the major milestones that enabled T cell immune monitoring [3].

With all the possible sources of damage to PBMCs prior to performing the actual test, T cell assays are inconceivable without proper controls to verify the functional fitness of these cells. Establishing the ratio of live/dead/apoptotic cells in the PBMC before testing them is helpful yet insufficient to identify their fitness. The functionality of antigen-specific T cells can only be established by measuring exactly that, which in turn requires positive control antigens, to which ideally all humans can be expected to have developed T cell immunity. This article is dedicated to the study of such positive control antigens.

In ELISPOT and the related FluoroSpot assays, antigen-specific T cells are visualized by detecting the cytokines they release following exposure to the test antigen: these cytokines are captured around each secreting cell on a membrane that has been pre-coated with cytokine-specific antibodies. Thus, the secretory footprint of each antigen-specific T cell is retained on the membrane in the form of a cytokine “spot”. The subsequent visualization of these plate-bound cytokine “spots” permits one to count the number of test antigen-specific T cells (expressed as “spot forming units” or SFU) present within all PBMC plated in a well. In this way, the frequency of antigen-specific T cells, and thus the magnitude of antigen-specific T cell immunity, can be established. Measurements of multiple cytokines simultaneously, either in double-color ELISPOT or multi-color FluoroSpot assays, can also define the effector lineage(s) of the antigen-specific T cells. In this study we focus on IFN-γ-producing type 1 cells, because in healthy subjects Th1 memory cells prevail by far, while Th2 and Th17 cells are induced only in low numbers by some antigens, and in a small subset of healthy subjects [4]. As is the case for any functional T cell assay, ELISPOT assays are also critically dependent on the functionality of the T cells and APC being preserved after storage/shipment of the blood, isolation of PBMC, and freeze-thawing of the cells before the actual test is performed [5].

A major limitation to T cell immune monitoring is that the choice of the antigen/peptide that is used in any functional T cell assay will define whether the memory T cells that have been induced in vivo will be detected at all. For immune monitoring with exogenous protein antigens this is not an issue, but such will largely detect only CD4+ T cells, and not CD8+ T cells [6]. When exogenous protein antigens are added to PBMCs, professional APCs (macrophages, dendritic cells and B cells) will process and present the antigen. The APCs uptake the protein antigen, degrade it, and load it onto MHC class II molecules (but not efficiently onto MHC class I molecules). The APC then transports the peptide-loaded class II molecule to its cell surface for antigen presentation to CD4+ T cells. When protein antigens are used for T cell recall assays, knowing the HLA-class II alleles expressed by the test subject, or the exact peptide epitope that is being presented to the CD4 cells is not important. Natural antigen processing and presentation mechanisms inherent to the APC select the relevant epitopes for each test subject without involving human best guessing. Equally importantly, when using protein antigens, no epitope will be left behind, but the full antigen-specific CD4+ T cell repertoire induced in vivo during the immune response will be detected in vitro in the recall assay.

Reliably detecting antigen-specific CD8+ T cells in PBMCs is much more intricate. CD8+ T cells evolved to survey ongoing protein synthesis within cells of the body, thus permitting CD8+ T cells to identify virally-infected or malignant cells in order to kill them. During protein synthesis within every cell, defective byproducts arise and are quickly degraded by the proteasome into peptide fragments. Some of these peptides are transported to the endoplasmic reticulum where they are loaded onto nascent HLA Class I molecules and such peptide-loaded class I molecules are transported to the cell surface where they are displayed for recognition by CD8+ T cells [7]. As the HLA gene complex is polygenic and highly polymorphic, and each allelic HLA class I molecule has a unique peptide binding specificity [8], this natural antigen presentation process results in a unique array of peptides presented in each individual, which is dictated by the unique HLA allele composition in said individual. To further complicate things, not all peptides presented will elicit a CD8+ T cell response in vivo: even HLA allele-matched individuals—who should present the same peptide epitopes when infected with the same virus—frequently develop aleatory CD8+ T cell response patterns in vivo for the individual epitopes [9].

Presently, the CEF peptide pool is the gold standard to test the functionality of antigen-specific T cells, and as such is typically included as the positive control in most T cell assays. It originally consisted of 23 well-defined CD8+ T cell epitopes of CMV, EBV, and Flu virus that have been selected to match HLA class I alleles that are frequent in Caucasians [10]. Aiming to detect CD8+ T cells in Caucasians only, the CEF peptides were not intended as a universal positive control. It was soon realized that these 23 peptides were insufficient to recall CD8+ T cells in most subjects, and the number of peptides was increased to 32; constituting the extended CEF pool [11]. There remains an urgent need for a universal positive control without any racial bias. In this communication, we focus on identifying such. There is also an urgent need for negative controls, in particular when using mega peptide pools that consist of hundreds of peptides. For the latter, we refer to a recent communication from our laboratory [12].

## 2. Materials and Methods

### 2.1. PBMC

Two hundred and ten randomly selected healthy human donors were obtained from the ePBMC library of Cellular Technology Limited (CTL, Shaker Heights, OH, USA, Cat# CTL-IP1). The HLA class I type, age, sex and race for these donors is specified in Appendix A. These donors were recruited by Hemacare (Van Nuys, CA, USA) and the PBMCs were isolated by leukapheresis at Hemacare using Hemacare’s IRBs. The PBMC were cryopreserved following protocols that maintain full T cell and APC functionality upon thawing [3], and were stored in liquid nitrogen vapor until testing. Thawing, washing, and counting of the cryopreserved cells were done as previously described. Within 2 h after thawing, the cells were transferred into the ImmunoSpot^®^ assay.

### 2.2. CD4 and CD8 Depletion of PBMC

CD4+ and CD8+ T cell subsets were depleted from PBMC using magnetic bead-based CD4 and CD8 negative selection kits (Stem Cell Technologies, Vancouver, Canada). The cell separations were performed according to the manufacturer’s instructions.

### 2.3. Positive Control Antigens

#### 2.3.1. CEF

A pool of 32 well-defined HLA class I -restricted epitopes of CMV, EBV, and Flu virus as defined in [11]. These peptides are 8–11 amino acids long and were selected to recall CD8+ T cells. They were from, and are available through, CTL (Catalog # CTL-CEF-002).

#### 2.3.2. CEFX

JPT Peptide Technologies, Berlin, Germany (Product Code: PM-CEFX) has a pool of 176 known peptide epitopes for a broad range of HLA sub-types (class I and class II) and different infectious agents, namely *Clostridium tetani*, Coxsackievirus B4, *Haemophilus influenza*, *Helicobacter pylori*, Human adenovirus 5, Human herpesvirus 1, Human herpesvirus 2, Human herpesvirus 3, Human herpesvirus 4, Human herpesvirus 5, Human herpesvirus 6, Human papillomavirus, JC polyomavirus, Measles virus, Rubella virus, *Toxoplasma gondii*, and Vaccinia virus. These peptides are 9–15 amino acids long and were selected to recall both CD4+ and CD8+ T cells. CEFX was tested at 1 μg/mL.

#### 2.3.3. CPI

Protein antigens of CMV, Parainfluenza and Influenza viruses are described in [13]. CPI was from, and is available through, CTL, Catalog #CTL-CPI-001. CPI was tested at 6.25 µg/mL.

#### 2.3.4. CERI

One hundred and twenty-four peptides of CMV, EBV, RSV, and Influenza virus were used. The individual peptides, 9 amino acids long, were selected based on peptide binding predictions for a broad range of HLA class I alleles expressed in all human races, and diverse ethnic subpopulations [14]. The rational for composing the CERI peptide pool is provided in Appendix A. CERI was from, and is available through, CTL (Catalog # CTL-CERI-300). CERI was tested at 1 μg/mL.

#### 2.3.5. Anti-CD3 

The anti-CD3 antibody was from Sigma-Aldrich, St. Louis, MO, USA (clone OKT3, catalog # SAB4700040-100UG). It was tested at 0.05 μg/mL.

### 2.4. Human Interferon-γ ImmunoSpot^®^ Assay

The human interferon-γ (IFN-γ) ImmunoSpot^®^ test kits were used from CTL (catalog# hIFNgp-1M/10), and the assay was performed according to the manufacturer’s protocols. In brief, the PVDF membrane was coated with the IFN-γ capture antibody overnight, and then washed. The antigens were plated at the specified concentrations in 100 μL/well. The PBMC were added at 300,000 cells per well in 100 μL CTL Test Medium, and the plates were gently tapped on each side to ensure even distribution of the cells. After a 24 h incubation at 37 °C in a humidified CO_2_ incubator, during which the IFN-γ produced by the antigen-stimulated T cells was captured, the cells were discarded, IFN-γ detection antibody was added, and the spot forming units (SFU) were detected via enzyme-catalyzed substrate precipitation. The plates were air-dried prior to analysis. The plates were analyzed using an ImmunoSpot^®^ S6 Ultimate Reader from CTL (Catalog# S6UTM12). The numbers of SFU were established using the ImmunoSpot^®^ Software’s (from CTL) SmartCount™ and Autogate™ functions [15] that permit user-independent objective counting of SFUs. Spot counts reported for the respective antigen-stimulated test conditions are means from triplicate wells, without the medium control subtracted.

### 2.5. Statistical Analysis of ImmunoSpot SFU Counts

As ImmunoSpot^®^ counts are normally distributed among replicate wells, the utilization of parametric statistics is suited for identifying positive responses [16]. Accordingly, an independent samples T-test was done comparing SFU counts in the triplicate antigen-containing wells vs. the SFU counts in the triplicate medium control wells. A *p*-value < 0.05 was considered as the cut-off for a significant SFU increase.

## 3. Results

### 3.1. CEF and CERI Recall CD8+ T Cells, CPI Recalls CD4+ T Cells, and CEFX Recalls Both

CD8+ T cells recognize 8–11 amino acid long peptides presented to them on HLA class I molecules [8]. As the peptide-binding grove of class I molecules is closed on both ends, it cannot accommodate longer peptides [17]. MHC class II molecules, in contrast, cannot efficiently bind and present such short peptides to CD4+ T cells [17]. As both the CEF and the CERI peptide pools contain peptides of 8–11 amino acid length, one would expect both to recall CD8+ T cells only. We performed cell depletion experiments to verify this assumption. As shown in Figure 1, CD8+ T cell-depleted PBMC fractions (PBMC-CD8) lost the CEF and CERI peptide pool-triggered recall response vs. the unseparated PBMC while CD4+ T cell depletion (PBMC-CD4), in contrast, had no such effect. These data suggest that the CEF- and CERI-triggered IFN-γ SFU are indeed produced by CD8+ T cells.

We also tested purified CD8+ T cells obtained from PBMCs by negative selection. CEF and CERI activated IFN-γ SFU in these purified CD8+ T cell fractions without the need to add APC (Appendix A), further supporting the notion that CD8+ T cells are responding to CEF and CERI, as CD8+ T cells are HLA class I-positive and they can serve as APCs to each other.

CD4+ T cells recognize protein antigens that are presented by specialized HLA class II positive APCs (primarily macrophages, B cells and dendritic cells), which internalize, process and present the antigen to the CD4+ T cells [18]. CPI consists of native viral proteins, and as such can be expected to be recognized by CD4+ T cells [18]. Cell separation experiments confirmed this notion: CD4+ T cell depletion from PBMC abrogated the CPI-triggered recall response, whereas CD8+ T cell depletion had no effect on it (Figure 1). When we tested purified CD4+ T cells (obtained from PBMC by negative selection), we found that, as expected, in the absence of APCs, CPI recall responses could only be detected if APC were added to the purified CD4+ T cells (data not shown). Therefore, CPI recalls CD4+ T cells. CPI consists of proteins that need to be internalized, processed, and presented by HLA-class II positive APCs. In contrast, the short peptides contained in CEF, CERI and CEFX can bind directly to HLA-molecules on the APC’s surface, and thus their presentation does not require active antigen processing. CPI also tests the functionality of the APC compartment. In a previous study [19], we established that macrophages, DCs and B cells all function as APCs for CD4+ T cells in ELISPOT recall assays, processing and presenting antigens. There are differences in the CD4+ T cell activation kinetics though, being faster for DCs (6 h) than B cells (24 h) with macrophages in between, but by 24 h all of these APCs induced maximal IFN-γ production in CD4+ T cells. Therefore, we can assume that each of these APC subsets contributes to a 24 h CD4+ T cell recall assay equally with the relative contribution of each being defined by their respective frequencies in the PBMCs.

CEFX has been designed to be a universal positive control for the recall of CD4+ and CD8+ T cells alike and accordingly consists of peptides 9 to 15 amino acids long that are suited for direct HLA class I and HLA class II molecule binding, without the need for additional antigen processing. When tested on the PBMC-CD4 and PBMC-CD8 cell fractions vs. the unseparated PBMC, an impairment was seen in both cell fractions, confirming a mixed recall of CD4+ and CD8+ T cells.

The data so far show that a single positive control might not suffice for the comprehensive assessment of PBMC fitness for T cell immune monitoring. CEF and CERI are candidates to test the functionality of antigen-specific CD8+ T cells, but do not serve to assess the functionality of the APC compartment. For testing the functionality of CD4+ T cells, and that of the antigen processing machinery, CPI is an ideal candidate. CEFX tests the functionality of CD4+ and CD8+ T cells, but owing to the 9–15 amino acid peptides that can be loaded directly into class I and class II molecules, it does not address the functionality of antigen processing.

### 3.2. CEF Fails as a Positive Control in 48% of Test Subjects

A positive control should ideally work for all test subjects. To verify whether this is the case for the CEF peptide pool, we tested all 210 healthy donors currently available in the ePBMC library. Standard 24 h IFN-γ ELISPOT assays were performed using 300,000 PBMCs per well. The raw data are shown in Appendix A, including medium control and CEF-triggered SFU counts and the class I HLA type, sex, age, and race of the test subjects. The CEF peptide pool-triggered SFU counts ranged between “too numerous to reliably count” (TNTC, >500 SFU/well) and zero. From the perspective of a positive control, we argue that qualifying results should exceed 50 SFU/300,000 PBMC, because such response magnitudes are commonly seen with individual peptides or antigens [20]. Of the 210 donors tested, 100 (48%) fell in the <50 SFU/300,000 PBMC category (Figure 2A). Forty-nine of the 210 donors (23%) showed no CEF response (<10 SFU/300,000 PBMC), and 51 subjects (24%) of all donors tested fell in the 10-49 SFU/300,000 PBMC category. The breakdown of CEF responses by the race of the test subjects is shown in Figure 2B.

### 3.3. CEF Non-Responder PBMCs Respond to Anti-CD3 Stimulation

Non-responsiveness to a positive control could mean that either the PBMCs are damaged/non-fit or the positive control itself is suboptimal. To distinguish between these two fundamentally different scenarios, we subjected samples of PBMCs from CEF non-responder subjects to anti-CD3 stimulation. As listed in Table 1, with few exceptions, anti-CD3 stimulation triggered vigorous IFN-γ SFU formation in CEF non-responder PBMCs. The only subject with an impaired PBMC response to anti-CD3 stimulation was ID 82. Reduced anti-CD3 triggered IFN-γ SFU formation using samples of ID 101, 118, and 131 also suggests that these PBMCs might qualify as impaired too. However, the strong anti-CD3 responsiveness of all other CEF non-responder PBMCs establishes that it is not these PBMC’s impaired functionality, but instead the insufficient formulation of the CEF peptide pool that accounted for the CEF-negative result.

### 3.4. CEF Non-/Low-Responder PBMCs Respond to CERI, CEFX and CPI Stimulation

To further assess the fitness of CEF-non/low-responsive PBMCs, we included CERI, CEFX, and CPI antigens into the functionality testing. Like anti-CD3, they too induced in part very high SFU numbers in the CEF non-responders (Table 1 and Figure 3C) and CEF low-responders as well (10–49 SFU, Figure 3F). These very strong antigen-specific CD8+ and CD4+ T cell recall responses in CEF-low/non-responding PBMCs further establish that it is not these PBMC’s impaired functionality, but the insufficient formulation of the CEF peptide pool itself that accounts for the CEF-negative results.

### 3.5. Low/Non-Responders to CERI, CEFX and CPI Are Rare

Encouraged by the above findings, we set out to test all 210 subjects’ PBMCs for recall responses to CERI, CEFX and CPI. As shown in Figure 4, only 4% of these PBMCs were negative for CERI, <1% for CEFX, and 2% for CPI, respectively, compared to the 23% CEF-negatives (Figure 1). In the borderline/low category (10–49 CEF-induced SFU/300,000 PBMCs), there were 13% for CERI, 10% for CEFX, and 5% for CPI, compared to the 24% of CEF-responders in this category. Genuine positive controls can be expected to induce a stronger recall response than individual antigens do, a threshold we empirically set at <50 SFU/300,000 PBMCs. For CEF, 52% (110/210) of the test subjects’ PBMCs reached this threshold (see Figure 1). In contrast, this threshold was reached for CPI by 93% (196/210), for CEFX by 89% (187/210), and for CERI by 83% (174/210) of the test subjects’ PBMCs.

## 4. Discussion

The data presented here suggest that the CEF peptide pool is a suboptimal positive control for testing the functionality of antigen-specific T cells in PBMCs. CERI, CEFX and CPI by far outperform CEF in this respect. However, each of these positive controls tests a different T cell compartment. CERI, like CEF itself, tests the functionality of antigen-specific CD8+ T cells. As the short peptides contained in the CEF and CERI pools bind directly to HLA class I molecules expressed on all cell lineages present in PBMCs, these peptide pools do not address the functionality of the antigen processing machinery. CPI, in contrast, consists of whole proteins that require antigen processing and presentation by professional APCs present within PBMCs, primarily macrophages, dendritic cells and B cells. Therefore, CPI tests for both CD4+ T cell and APC functionality. CEFX, consisting of peptides capable of direct binding to HLA class I and class II molecules, probes antigen-specific CD8+ and CD4+ T cell functionality; however, it is unable to address the functionality of the antigen processing compartment.

Viewing the data in Table 1, one might conclude that anti-CD3 is the ideal positive control for assessing PBMC fitness. Indeed, being a polyclonal CD4+ and CD8+ T cell stimulator, anti-CD3 activates in vivo differentiated type 1 (IFN-γ-producing) T cells in much higher frequency than individual antigens or pools of antigens can do. However, anti-CD3 antibodies result in unnatural T cell stimulation [21], which is fundamentally different from the serial triggering involved in regular T cell activation, during which the T cell receptor (TCR) oscillates between low-affinity binding to, and dissociation from, its ligand, the HLA-nominal peptide complex [22]. Therefore, in these authors’ eyes, in addition to anti-CD3, T cell immune monitoring is also in need of antigens that are suited to test physiologic T cell activation assessing the different requirements of CD4+ or CD8+ T cell detection in PBMCs, including the APC’s functionality.

The data presented here show how difficult it is to come up with universal positive controls for T cell immune monitoring. CPI came closest to detect antigen-specific T cells in all donors tested, inducing >50 SFU/300,000 PBMC in 93% of them. However, the response magnitude to CPI (and to all of the other positive control antigens tested) showed a wide span of inter-individual variations. This outcome is expected, as not all individuals have developed T cell immunity to all antigens contained in the respective positive control (in the case of CPI, CD4 responses to proteins from CMV, Parainfluenza, and Influenza viruses), and if they did, the frequencies of T cells targeting each of these viruses will differ among individuals dependent upon their immune status vs. the respective virus. Therefore, the inclusion of antigens from additional viruses into the CPI panel might be required to elicit larger recall responses in the remainder (7%) of CPI low/non-responders.

When CPI reactivity is tested, the viral proteins are processed and presented by the autologous APC present in the PBMC. Therefore, the correct and complete selection of epitopes displayed to CD4+ T cells on APCs is not a limiting factor; it occurs naturally. Thus, it can be assumed that the ex vivo recall with CPI assesses the entire in vivo primed CPI-specific CD4+ T cell repertoire. However, when peptides are used for recall, as in the case of CEF, CERI and CEFX, the peptides used for recall are not likely to address the entire virus-specific T cell repertoire.

The underlying assumption for creating the CEF pool was that there is immune dominant recognition of a few epitopes from CMV, EBV and influenza viruses. For example, it was assumed that a single peptide of CMV, pp65_495–503_, is immune dominant in all HLA-A*02:01-positive subjects and therefore this peptide would suffice to detect CMV-specific CD8+ T cells in all CMV-infected, HLA-A*02:01-positive humans [10]. In a recent study, complete epitope mapping was performed for the CMV pp65 protein on four CMV-positive, HLA-A*02:01-positive test subjects [23]. Peptide pp65_495–503_ was indeed immune dominant in one of these donors, but it was cryptic (it induced a borderline/low recall response) in another HLA-A*02:01-positive, CMV-positive donor, who in turn exhibited dominant CD8+ T cell recognition of two alternative pp65-derived epitopes. In yet two other HLA-A*02:01-positive, CMV-positive donors, pp65_495–503_ was subdominant, recalling low frequency SFU compared to other dominant epitopes of the pp65 antigen. Therefore, upon closer examination, the immune dominance of pp65_495–503_ does not hold up for HLA-A02:01-positive subjects. Rather pp65_495–503_ is just one of several potential CD8+ T cell epitopes to which CD8+ T cells respond in an apparently aleatory (dice-like) manner [9]. We confirmed this notion in a follow up study testing 10 additional HLA-A*02:01-positive subjects [12].

The above conclusion was also supported by a study [20] in which the individual CEF peptides, including pp65_495–503_, were systematically tested on high-resolution HLA-typed healthy test subjects. It was found that of 241 expected positive recall responses, in only 36 (15%) instances did the expected individual CEF peptides indeed recall strongly positive (dominant) CD8+ T cell responses. In 41 (17%) instances they induced low frequency CD8+ T cells (subdominant), and in 68% of the test cases, these expectedly immune dominant epitopes recalled a borderline or negative (cryptic) CD8+ T cell response. Similar results were obtained for the CMV pp65_495–503_ peptide in HLA-A*02:01-positive, CMV-infected donors in the aforementioned study [20]. This observation was confirmed in a follow-up study that involved 52 HLA-A*02:01-positive, CMV-infected subjects: even though all these subjects developed strong T cell immunity to CMV, 8% of them either did not respond to the CMV pp65_495–503_ peptide at all, and 27% displayed a subdominant/cryptic response to the pp65_495–503_ peptide [12].

Immune dominance of single epitopes might therefore be the exception even in HLA-allele matched humans, and aleatory recognition of multiple epitopes the rule [9], suggesting that positive controls that rely on a few select peptides are prone to underestimate, or outright miss the virus-specific memory T cells they were designed to detect. The 32 peptide-containing CEF pool contains only five CMV peptides attempting to detect CMV-specific CD8+ T cells across the human population. The shortcoming of the CEF peptide pool is therefore linked to the absence of immune dominance. Thus, due to the tremendous HLA class I allele diversity in the human population, these 32 peptides of CMV, EBV and Flu virus are insufficient to reliably detect CD8+ T cells specific for these viruses. Relying on 176 known epitopes of a larger diversity of viruses, and 124 peptides of predicted epitopes, the CEFX and CERI peptide pools come much closer, but still do not completely meet the goal of all-encompassing positive controls. The long-term solution will be to further increase the number of antigens and peptides in positive controls. The short-term solution might be to use all three highly improved positive control antigens presently available: CEFX, CERI and CPI, in addition to anti-CD3. Due to the response magnitudes elicited by these, it is not necessary to test each in replicates, and therefore all three positive control antigens can be tested with the same numbers of PBMCs as presently done when CEF is tested in triplicates. Testing of all three positive controls is not only advisable because they are complementary in covering a wider spectrum of recall antigens, to each of which test subjects are prone to have various levels of T cell immunity, but also because in this way the functionality of CD4+, CD8+ T cells, and of the APC are separately addressed, which makes a comprehensive assessment of PBMC fitness possible.

## Figures and Tables

**Figure 1 cells-10-00248-f001:**
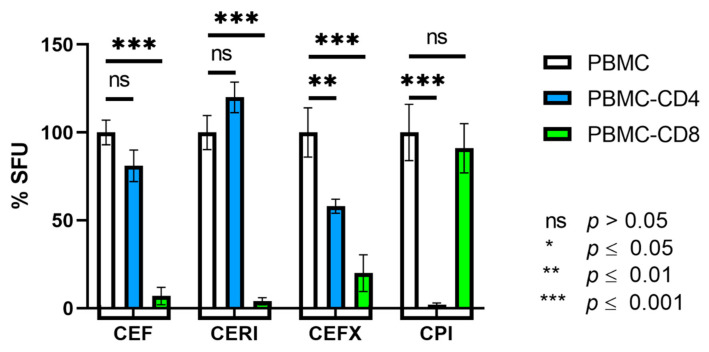
Identifying the CD4/CD8 lineage of T cells responding to antigens CEF, CERI, CEFX, and CPI. The SFU counts in the unseparated PBMC was set as 100%, to which the SFU counts in the CD4 cell-depleted PBMC (PBMC-CD4 that still contain the CD8+ T cells) and the CD8 cell-depleted PBMC (PBMC-CD8 that sill contain the CD4+ T cells) are compared. PBMC, PBMC-CD4, and PBMC-CD8 were all adjusted to 300,000 cells/well. The PBMC tested were from Donor ID 162 and the 100% SFU counts for the CEF, CERI, CEFX, and CPI-responses in the unseparated PBMC were 195, 266, 400, 486, SFU, respectively. For each experimental condition, means and standard deviations calculated from triplicate wells are shown, and an independent samples student’s *T*-test was done to detect significant differences between the specified conditions.

**Figure 2 cells-10-00248-f002:**
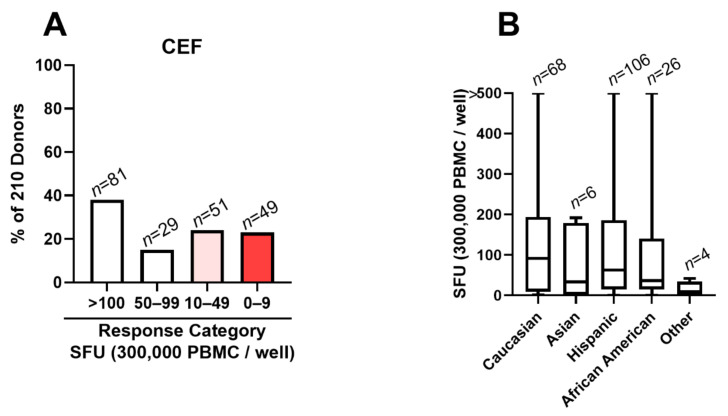
CEF peptide pool-triggered recall response in 210 healthy human donors. A standard 24 h IFN-γ ImmunoSpot^®^ assay was performed testing the CEF peptide pool induced SFU numbers. (**A**) Response magnitudes have been divided into the specified SFU categories. Representative wells for these response categories are shown in Appendix A. The percentage of subjects falling in each response category is shown on the *Y*-axis, and the number of subjects in each category is specified above the bars. (**B**). The breakdown of CEF responses by race is shown as a box and whiskers plot.

**Figure 3 cells-10-00248-f003:**
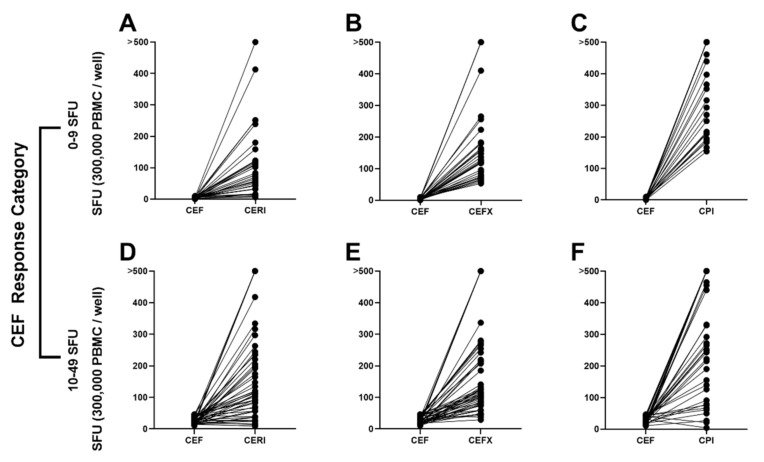
CERI-, CEFX-, and CPI-induced recall responses in CEF non-/low-responder PBMC. (**A**–**C**) Subjects whose PBMCs fell with <10 SFU/300,000 PBMCs into the CEF non-responder category (n = 49), or (**D**–**E**) with 10–49 SFU/300,000 PBMCs into the CEF low-category (n = 51) were tested in a standard 24 h IFN-γ ImmunoSpot^®^ assay for CERI (**A**,**D**), CEF-X (**B**,**E**) and CPI (**C**,**F**) recall at 300,000 PBMCs/well. SFU counts of the same PBMC are connected with a line.

**Figure 4 cells-10-00248-f004:**
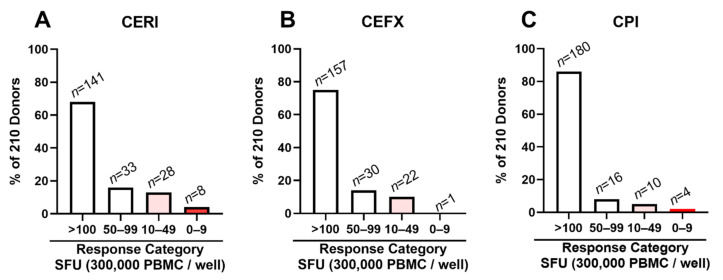
CERI (**A**), CEFX (**B**) and CPI (**C**)-triggered recall responses in 210 healthy human donors. A standard 24 h IFN-γ ImmunoSpot^®^ assay was performed testing the antigen-induced SFU numbers. Response magnitudes have been divided into the specified response categories. The percentage of subjects falling into each response category is shown on the *Y*-axis, and the number of subjects in each category is specified above the bars.

**Table 1 cells-10-00248-t001:** CEF-negative PBMC can respond to other T cell stimuli. PBMC that showed no response to CEF (<10 SFU/300,000 PBMC) were tested simultaneously for CERI, CEFX, CPI and anti-CD3-induced T cell activation in a standard 24 h IFN-γ ImmunoSpot^®^ assay at 300,000 PBMC/well, in triplicates each. Mean SFU counts are shown for all conditions

Subject	Media	CEF	CERI	CEFX	CPI	Anti-CD3
ID 30	1	8	117	96	212	>500
ID 51	2	7	138	41	268	>500
ID 68	2	2	413	265	270	>500
ID 69	0	0	2	4	6	436
ID 71	2	4	51	160	352	>500
ID 82	1	1	15	27	4	129
ID 87	2	1	48	17	158	>500
ID 98	2	3	252	118	250	>500
ID 101	1	6	72	29	29	233
ID 103	1	1	112	180	194	>500
ID 118	0	2	32	18	212	243
ID 123	2	4	180	58	366	348
ID 128	1	7	121	53	183	459
ID 131	1	9	47	23	37	260
ID 133	3	2	159	89	316	>500
ID 138	1	5	32	123	461	>500
ID 144	0	7	55	127	166	>500
ID 147	1	5	77	156	>500	>500
ID 150	0	8	83	410	216	>500
ID 160	1	2	12	43	27	495
ID 168	3	6	33	136	439	>500
ID 169	2	8	102	70	293	381
ID 191	1	0	70	75	154	403
ID 196	1	5	32	20	154	>500

## Data Availability

Data is contained within the article and or Appendix A.

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
