# Peer review of "CERI, CEFX, and CPI: Largely Improved Positive Controls for Testing Antigen-Specific T Cell Function in PBMC Compared to CEF"

_cells, 2021, doi:10.3390/cells10020248_

Round 1
Reviewer 1 Report
Brief Summary
The soundness of T cell immune monitoring studies hinges on the quality of PBMCs that are isolated for ex vivo testing. Test of quality relies mainly on functional recall assays which make use of control peptides. In the present study the authors aimed to provide more rigorous and reliable control peptide pools capable of recalling memory CD4+ and CD8+ T cell responses. Authors demonstrate that whereas the widely used CEF peptide pool is defective in T cell recall responses in most test PBMCs, CEFX, CERI, and CPI peptide pools elicit improved recall responses and as such should be used in tandem with CEF for better outcomes. The scope of the study is interesting. However, authors must address the following comments before the manuscript is considered appropriate for publication.
Comments
- In their experiments, authors investigated response to the various peptide pools using IFN-g Since the study investigates both CD4+ and CD8+ T cell recall response. It will be worthwhile conducting ELISPOT assays for multiple cytokines (e.g., IL-2, IL-4 etc.) if possible, in order to increase the confidence on the appropriateness of these peptide pools as positive controls.
- Authors need to analyze data in Figure 1 and indicate if there are any statistical differences in the observed patterns within the different peptide treatments. For example, for CEF treatment, indicate any statistical significance between PBMC, PBMC-CD4, and PBMC-CD8.
- In connection to comment 2 above, authors need to discuss the elevated levels of IFN-g in the PBMC-CD4 group treated with CERI if the difference in IFN-g level between unseparated PBMC and PBMC-CD4 is statistically significant.
- Figure 2 shows that almost half of the individual PBMC responses to CEF were below the 50 SFU/300000 PBMC threshold. I would therefore want to know which PBMCs were used for experiments in Figure 1 (from a single responder individual or a pool of responder individuals?), and what quantum of SFU counts in the unseparated PBMCs was used as the 100% standard.
- Since CPI efficiently tests functionality of both the T cell and APC compartments, it would be interesting to ascertain which APC subset (macrophages, dendritic cells, B cells) is functionally relevant in the processing and presentation of CPI peptides to the T cells.
- Authors may also need to show by flow cytometry the CD4+ and CD8+ populations in the PBMC pool before and after treatment with the control peptides.
- Isn’t the number of CPI responders 196 instead of 195 in line 300, based on the numbers in Figure 4C.
- Title of Section 3.1 (line 201) is cryptic. It needs to be rephrased.
- Correct typographical errors (e.g., Supplemental Table 1). It would be better to fix it to “Characteristics of Subjects Tested”.
Author Response
- In their experiments, authors investigated response to the various peptide pools using IFN-g Since the study investigates both CD4+ and CD8+ T cell recall response. It will be worthwhile conducting ELISPOT assays for multiple cytokines (e.g., IL-2, IL-4 etc.) if possible, in order to increase the confidence on the appropriateness of these peptide pools as positive controls.
Answer 2. In our publication https://doi.org/10.3390/cells6030029 we established for 12 common recall antigens the T cell cytokine signatures for a cohort of healthy human donors. The data showed that these antigens elicited IFN-g producing type 1 T cells but rarely other cytokines (IL-4, IL-5, IL-17) and those were induced only by select antigens in select subjects (summarized in Table 1 of that publication). This is because in healthy donors type 1 T cells prevail by far over type 2 or Th17, the latter being induced only by certain antigens, in certain donors (e.g. atopic individuals) and in the context of certain pathologies (e.g. allergies). Therefore, the detection of such T cell subsets (Th2, Th17) does not lend itself to a general T cell functionality test. This is an important point raised by the Reviewer, and in response, we commented on it in the revised manuscript (lines 56-58).
- Authors need to analyze data in Figure 1 and indicate if there are any statistical differences in the observed patterns within the different peptide treatments. For example, for CEF treatment, indicate any statistical significance between PBMC, PBMC-CD4, and PBMC-CD8.
Answer 3. We apologize for not having described this figure in sufficient detail. The antigens have been tested each in triplicate wells on PBMC vs. the CD4+ or CD8+ T cell-deleted subsets thereof; data are shown for a representative donor. In response to the Reviewer’s valid critique, in the revised Figure 1 we now show the statistical analysis of the data and included the missing information into the legend of Figure 1.
- In connection to comment 2 above, authors need to discuss the elevated levels of IFN-g in the PBMC-CD4 group treated with CERI if the difference in IFN-g level between unseparated PBMC and PBMC-CD4 is statistically significant.
Answer 4. Introducing the statistical analysis of these data it can now be seen that this slight elevation is not statistically significant.
- Figure 2 shows that almost half of the individual PBMC responses to CEF were below the 50 SFU/300000 PBMC threshold. I would therefore want to know which PBMCs were used for experiments in Figure 1 (from a single responder individual or a pool of responder individuals?), and what quantum of SFU counts in the unseparated PBMCs was used as the 100% standard.
Answer 5. The Reviewer is correct, we should have provided this information. We included the revised Legend of Figure 1.
- Since CPI efficiently tests functionality of both the T cell and APC compartments, it would be interesting to ascertain which APC subset (macrophages, dendritic cells, B cells) is functionally relevant in the processing and presentation of CPI peptides to the T cells.
Answer 6. In a previous study (PMID: 17632036) we have established that macrophages, DC and B cells all function as APC for CD4+ T cells in ELISPOT recall assays processing and presenting antigens. There are differences in the CD4+ T cell activation kinetics though, being faster for DC (6h) than B cells (24h) with macrophages in between, but by 24 h all of these APC induced maximal IFN-g production in CD4 T cells. Therefore, we can assume that each of these APC subsets contributes to a 24 h CD4+ T cell recall assay equally with the relative contribution of each being defined by their respective frequencies in the PBMC. We thank the Reviewer for bringing up this issue, we now comment on it in lines 184-193 of the revised manuscript.
- Authors may also need to show by flow cytometry the CD4+ and CD8+ populations in the PBMC pool before and after treatment with the control peptides.
Answer 7. We had stringent internal controls for such cell depletion experiments:
- CEF is a well-established peptide pool consisting of previously defined short class I-restricted epitopes. As expected, CD8+ T cell depletion close to completely abrogated the CEF-triggered IFN-g- production, not only reproducing that the responding cells were indeed CD8+, but also proving that we succeeded with the CD8 cell depletion.
- CPI consists of protein antigens that need to be processed by APC for CD4+ T cell stimulation. It was previously established that CERI elicits CD4 cells (doi:10.3390/cells6040047). In Fig1, CD4 cells depletion close to completely abolished the recall response to CPI. This result therefore not only reproduces that the CERI-reactive IFN-g-producing cells are indeed CD4+, but also provides the internal control that we succeeded with the CD4+ cell depletion.
- CERI consists of 9 aa long peptides selected for class I binding. Nine aa long peptides are in general too short to constitute class 2-restricted epitopes. The same CD8- depleted cell fraction that was depleted of CEF reactivity also lacked reactivity to CERI, but the CD4 cell depletion had no significant effect on the CERI response, clearly establishing the CERI-reactive T cells are of the (expected) CD8+ T cell class.
- CEFX consists of 9-15 amino acid long peptides that (according to the manufacturer “have been selected to recall both CD4+ and CD8+ T cells”. This (expected) outcome was indeed seen in our internally controlled cell depletion experiments.
- Following Reviewer 2’s suggestion we added Supplemental Figure 2, which we just mentioned in the original submission as “data not shown”, now showing that purified CD8+ T cells can present CEF, CERI and CEFX peptides to each other in the absence of other APC, but purified CD4+ cells (lacking class II positive APC), cannot present to each other any of these antigens. These data are also in line with the well-established antigen presentation requirements for CD4+ and CD8+ cells.
We got only 10 days to resubmit this manuscript. This time frame is too short to repeat this cell separation experiment including flow cytometry controls and we think that is also not needed as the data are very clear, are internally controlled, and provided the expected results.
- Isn’t the number of CPI responders 196 instead of 195 in line 300, based on the numbers in Figure 4C.
Answer 8. The Reviewer is right, the correct number is 196, not 195. Thanking the Reviewer, we have made the correction.
- Title of Section 3.1 (line 201) is cryptic. It needs to be rephrased.
Answer 9. We thank the reviewer for this feedback. We rephrased it to: “CEF and CERI Recall CD8+ T Cells, CPI Recalls CD4+ T Cells, and CEFX Recalls Both” (line 152 in the revised manuscript).
- Supplemental Table 1) rename to “Characteristics of Subjects Tested”.
Answer 10. Thank you for the suggestion, we have adopted the suggested title for Supplemental Table 1.
- Correct typographical errors
Answer 11: we thoroughly proof-read the manuscript again for typographical errors and checked the document using “Grammarly”.
We thank the Reviewer for the thorough critique. We attempted to address each point and by doing so we hope to have improved the clarity of the presentation including that of the research design.
Reviewer 2 Report
The manuscript “CERI, CEFX, and CPI: largely improved positive 2 controls for testing antigen-specific T cell function in 3 PBMC compared to CEF” by Lehmann et al presents a study evaluating technical aspects, mainly positive control assessments, of restimulation experiments using peripheral blood derived mononuclear cells (PBMCs). Using an extended pool of virus specific restimulation peptide mixes representing different viral epitopes (CEF, CERI, CPI, CEFX), the authors demonstrate that depending on the T cell subset to be evaluated, different peptide mixed are needed. Indeed, CEF, a peptide mix commonly used, fails as a positive control in half of the PBMC samples tested, while these cells still respond to polyclonal CD3 stimulation, thus delineating CEF as inapt to serve as a wide-spanning positive control. In contrast, using CERI, CEFX and CPI stimulation, the authors demonstrate a high frequency of positive reponders in the CEF non-responder population, comparable to antigen-independent anti-CD3 stimulation. Together, the authors conclude that the CEF peptide pool commonly used as a positive control to assess PBMC viability is a suboptimal positive control for testing viability and functionality of T cells in PBMC, while CERI, CPI and CEFX together with anti-CD3 stimulation represent more suitable positive controls to assess both viability and functionality of ex vivo PBMC preparations.
General critique:
The study presented by Lehmann et al presents important insights into the assessment of PBMC viability and function using positive controls representing different aspects of APC, CD4 and CD8 T cell function. Using different epitope mixtures representing speciifc aspects of CD3 T cell and APC biology together with polyclonal CD3 stimulation, they guide the development of future positive controls in human PBMC studies. Thus, I consider this manuscript of high relevance to a broad readership of both experimentally focused and translationally minded scientists, assessing PBMC responses to self and foreign antigens. I thus recommend publication of this important study, if a few minor points are addressed in a revised version of the manuscript:
- While both introduction and discussion are quite instructive, I still recommend shortening both parts to focus on the aspects relevant to this study.
- Figure 1: label nomenclature is misleading. Please adjust to clarify that PBMC-CD4 is actually CD4 depleted of CD8 etc.
- Please also show the enrichment experiment mentioned in context with Figure 1, e. g. as supplementary figure.
- While the experiments presented here assess the relevance of viral peptides and CD3 stimulation as positive controls, is there a good experimental system to be included as a negative control in these studies?
Author Response
- While both the introduction and discussion are quite instructive, I still recommend shortening both parts to focus on the aspects relevant to this study.
Answer 11. Thanking for this suggestion, we have shortened the manuscript.
- Figure 1: label nomenclature is misleading. Please adjust to clarify that PBMC-CD4 is actually CD4 depleted of CD8 etc.
Answer 12. Thanking the Reviewer for bringing it to our attention, we now revised the legend of Figure 1 to clarify this point.
- Please also show the enrichment experiment mentioned in context with Figure 1, e. g. as supplementary figure.
Answer 13. Following the Reviewer’s suggestion, we introduced it as Supplemental Figure 2.
- While the experiments presented here assess the relevance of viral peptides and CD3 stimulation as positive controls, is there a good experimental system to be included as a negative control in these studies?
Answer 14. This is a very important point, in particular when it comes to using mega peptide pools for T cell immune monitoring. A manuscript we just submitted (already accessible as pre-print doi: https://doi.org/10.1101/2020.11.29.402677 ) is in large dedicated to this issue. We now refer to it in the revised manuscript (lines 95-98).
We truly appreciate and thank this Reviewer for the positive feedback.
Reviewer 3 Report
The manuscript submitted by Lehmann et al., entitled as “CERI, CEFX, and CPI: largely improved positive controls for testing antigen-specific T cell function in PBMC compared to CEF” details comparison of T cell response against different protein antigens. These tests and reagents are helpful in the basic assessment of the PBMC functional fitness. This topic is of immense interest.
The manuscript is not prepared well. Methods and results should include more details, as pointed below. Further, authors noted that CEF couldn’t be relied as a gold standard, in general. However, this reviewer does not agree with this notion. Authors are presenting their study in a very subjective manner. CEF peptides were designed to test among Caucasian individuals with prior exposure to CMV, EBV, and Flu. As stated in reference #8 of the manuscript, "The chosen epitopes are presented by the most common Caucasian HLA types, whose cumulative frequencies represent >100% of Caucasian individuals (Imanishi et al., 1992). The use of the peptides should theoretically enable us to examine CD8+ T cell responses in the majority of Caucasians previously exposed to CMV, Flu and EBV.” Since test subjects in the present study are mixed race, they are bound to fail in the CEF based tests. Authors need to describe the HLA typing, allele frequency, and racial bias in HLA dependent manner and should reconsider their conclusions. This reviewer suggests that authors should rather emphasize that CEF is not a "universal control" and that their study is aimed at identifying "A universal control" without any racial bias.
The introduction section is very lengthy and provides book knowledge. It can be concise and precise, with references to the review articles or other past researches.
Material and methods section 2.1, since authors analyzed CD4 T cell response, HLA class II typing results should also be shown in supplementary table 1. Authors should also include allele frequency. Regarding different pools, CEF, CEFX, CPI, and CERI, authors should emphasize that what allele frequency of individual racial groups studied here (caucasia, Asian, African American, Hispanic) do these pools cover. How many peptides are overlapping in different groups? A table summarizing HLA class, viral groups covered, length of peptides, and covered allele frequency of individual race group will be very helpful.
Fig.1 Statistical analysis is not shown? How many subjects were tested? Labeling should be corrected to something like “PBMC (-CD4)”. Data for purity of cells, e.g. flow cytometry, should be provided. Data comparing unseparated PBMC response to different peptides should also be shown.
Line 228-29, states that CPI tests APC processing and presentation of the antigen. It is not clear, why processed antigen cannot be presented to the CD8 T cells?
Fig.2, data should be presented as “Box and Whiskers plot, with all data point shown”. It will give a comprehensive idea of breadth of the response. For 2A, representative ELISPOT wells should be shown to give visuals of SFU counts in different categories.
Fig. 4. Data should be shown in comparison to CEF. One way could be to include SFU counts against various pools in the supplementary table 1.
The conclusion that all three proposed pools should be used to assess PBMC fitness is not practical. Authors need to consider time and efforts in a clinical setting.
Font size and types are varied throughout the manuscript, and should be corrected.
Author Response
The manuscript submitted by Lehmann et al., entitled as “CERI, CEFX, and CPI: largely improved positive controls for testing antigen-specific T cell function in PBMC compared to CEF” details comparison of T cell response against different protein antigens. These tests and reagents are helpful in the basic assessment of the PBMC functional fitness. This topic is of immense interest.
The manuscript is not prepared well. Methods and results should include more details, as pointed below. Further, authors noted that CEF couldn’t be relied as a gold standard, in general. However, this reviewer does not agree with this notion. Authors are presenting their study in a very subjective manner. CEF peptides were designed to test among Caucasian individuals with prior exposure to CMV, EBV, and Flu. As stated in reference #8 of the manuscript, "The chosen epitopes are presented by the most common Caucasian HLA types, whose cumulative frequencies represent >100% of Caucasian individuals (Imanishi et al., 1992). The use of the peptides should theoretically enable us to examine CD8+ T cell responses in the majority of Caucasians previously exposed to CMV, Flu and EBV.” Since test subjects in the present study are mixed race, they are bound to fail in the CEF based tests. Authors need to describe the HLA typing, allele frequency, and racial bias in HLA dependent manner and should reconsider their conclusions. This reviewer suggests that authors should rather emphasize that CEF is not a "universal control" and that their study is aimed at identifying "A universal control" without any racial bias.
Answer 15. We gratefully accept the Reviewers suggestion to adopt the wording “CEF is not a "universal control" and that we aimed at identifying "a universal control" without any racial bias.” We introduced this language in the Abstract, and in the Introduction (lines 92-96) and deleted from the introduction passages that call into question CEF’s suitability as a positive control for Caucasians.
(We did so not because we agree with the Reviewer, but because it is a side issue for this manuscript. First, we wish to comment on the above citation: “The use of the (23 CEF) peptides should theoretically enable us to examine CD8+ T cell responses in the majority of Caucasians previously exposed to CMV, Flu and EBV.” As can be seen in or Figure 2B, about 30% of Caucasians give either no or borderline responses even to the extended 32 peptide CEF pool. We did not test it systematically, but for several of such CEF-non-responder Caucasian donors we found vigorous recall responses to 15-mer peptide pools to these three viruses. These results clearly show that the lack of CEF response in these Caucasian donors did not result from them not having been infected by these three viruses, or having developed week immunity to these viruses, but resulted from the CEF peptide pool not containing the actual peptides that the CD8+ T cells in these subjects target. If the Reviewer wishes, we can add a Supplemental table with such data that clearly establishes that CEF peptides do not necessarily provide sufficient epitope coverage even for Caucasians. This outcome is in line with our Answer 19 in which we address how unreliable predicted epitope dominance patterns are. Still, we gladly adopt the Reviewer’s wording (CEF is not a "universal control" and that we aimed at identifying "a universal control" without any racial bias) and deleted from the introduction language that calls CEF into question. This is because our intent is not to badmouth CEF, but merely to draw attention that better positive controls are now out there for all races, including Caucasians).
Answer 16. As to the Reviewer’s point “. Authors need to describe the HLA typing, allele frequency, and racial bias in HLA dependent manner and should reconsider their conclusions” please see our Answer 19, below.
The introduction section is very lengthy and provides book knowledge. It can be concise and precise, with references to the review articles or other past researches.
Answer 17. Following the Reviewer’s advice, we substantially shortened the introduction.
Material and methods section 2.1, since authors analyzed CD4 T cell response, HLA class II typing results should also be shown in supplementary table 1. Authors should also include allele frequency.
Answer 18. Following the Reviewer’s advice, the HLA class II alleles have now been introduced into the revised Supplementary Table 1.
Answer 18. Regarding allele frequencies, please see Answer 19.
Regarding different pools, CEF, CEFX, CPI, and CERI, authors should emphasize that what allele frequency of individual racial groups studied here (Caucasian, Asian, African American, Hispanic) do these pools cover. How many peptides are overlapping in different groups? A table summarizing HLA class, viral groups covered, length of peptides, and covered allele frequency of individual race group will be very helpful.
Answer 19. We agree with the Reviewer that these are interesting questions but we think that the CERI and CEFX peptide pools we tested here (each consisting of over 100 peptides) are way too complex to provide insights into the actual restriction element utilization and its predictability. Testing the individual peptides singly, in contrast, permits to directly address this issue. In our publication PMID: 27108305 we did exactly that for the 32 individual CEF peptides. We found that of 179 responses elicited by the individual CEF peptides, 68 % were matching the prediction but 32 % were not (i.e, as far as predictions go, 32 % were false positive, with the peptide being restricted by other than the expected allele). We also found that of predicted 241 immunodominant recall responses, in only 77 instances (32%) were dominant or subdominant recall responses actually detectable (i.e., 68% false negative predictions) whereas such donors responded to other peptides of the same virus. Studying Hepatis C (10.1006/clim.2001.5193 ) we also found that previously defined and in silico predicted epitopes close to completely failed to identify the actually targeted CD8+ T cell epitopes. In a recent study, presently under review, we looked at the highest possible resolution at the CD8+ T cell response to the HCMV pp65 antigen coming to the same conclusion: there is a fundamental “Discordance between the predicted vs. the actually recognized CD8+ T cell epitopes of HCMV pp65 antigen and aleatory epitope dominance” https://doi.org/10.1101/2020.11.06.371633 . We are not the only ones who call predictable epitope dominance in question ( 10.1016/j.immuni.2006.09.005 ). When talking to peers about unpredictable epitope recognition they are divided into two camps. About half of them comment “what else, that is in line with the very nature of the T cell response, and we saw and published on similar findings”). The other half feels alienated by their views and work over years being challenged. Either way. this issue cannot be resolved by studying pools containing over 100 peptides, they can only be resolved by testing the individual peptides vs. each donor’s HLA-type, as we did in the above studies. We did it for CEF, we could do it for CERI (which would warrant a paper on its own, however), but we cannot do it for CEFX because the composition of that mega peptide pool is held proprietary by JPT (yet CEFX is commercially available). CPI consists of proteins and therefore CD4+ T cell epitope selection occurs through natural antigen processing.
To clarify, we did select the CERI peptides per predicted binding to cover HLA alleles across the races. In part complying with the Reviewer’s suggestion to show allele frequencies in races, we added Supplemental Table 2 that lays out our strategy for the selection of the CERI peptides. However, in our view such epitope predictions just permit to narrow down the potential epitope space – I personally would be surprised if the actual epitope utilization for CERI and CEFX would differ from what we found for CEF, HepC, and pp95... I also do not expect the 100 plus peptides in CERI and CEFX to cover the expressed immunome of these viruses in all races.
Following the Reviewer’s suggestion, we deleted in the introduction passages that express our sceptic view on predictable epitope utilization for CEF and in general, leaving that issue to continued in-depth studies dedicated to this issue. Because these mega peptide pool data are not suited to resolve it, we seek the Reviewer’s consent to keep this manuscript focused on the practical suitability of the peptide pools as universal positive controls, being the only claim we make; a message Reviewer 3 judged “a topic of immense interest”.
Fig.1 Statistical analysis is not shown? How many subjects were tested? Labeling should be corrected to something like “PBMC (-CD4)”. Data for purity of cells, e.g. flow cytometry, should be provided. Data comparing unseparated PBMC response to different peptides should also be shown.
Answer 20. As the same questions were raised by Reviewer 1, we refer to our Answers 3-7 above. We hope the additional information provided there, along with the changes made in Figure 1 itself, its legend, and the text address the Reviewer’s questions.
Line 228-29, states that CPI tests APC processing and presentation of the antigen. It is not clear, why processed antigen cannot be presented to the CD8 T cells?
Answer 21. CPI consists of proteins, not peptides. Only rare DC subpopulations are capable of cross presentation (DOI: 10.1182/blood-2012-06-435644 ), too rare among class II positive APC in the blood to make out a significant compartment. We tested it over the years for “dozens” of protein antigens finding what we show in figure 1: once CD4 cells are depleted from PBMC, the residual cells (containing CD8+ T cells and plasmacytoid DC) are no longer stimulated by protein antigens, i.e., cross presentation does not play a detectable role in PBMC-based Elispot assays. As an example, please see Figure 3 in DOI: 10.3390/v7082828 . In that case, we tested for HCMV. Although there were plentiful CD8+T cells present in such donors, CD4 cell depletion completely abrogated the ability of CD8+ T cell to respond to HCMV protein antigens, establishing that cross presentation did not play a role significant enough for CD8+ T cells to become detectable when protein antigens are used in ELISPOT assays. We rephrased this statement to make this clearer.
Fig.2, data should be presented as “Box and Whiskers plot, with all data point shown”. It will give a comprehensive idea of breadth of the response. For 2A, representative ELISPOT wells should be shown to give visuals of SFU counts in different categories.
Answer 22. We thank the Reviewer for the “Box and Whiskers” plot suggestion. Figure 2B has been revised accordingly.
Answer 23. Following the Reviewer’s suggestion, we now show representative wells in the newly added Supplemental Figure 1.
Fig. 4. Data should be shown in comparison to CEF. One way could be to include SFU counts against various pools in the supplementary table 1.
Answer 24. Following the Reviewer’s suggestion, the data for CERI, CEFX and CPI are now also included in the revised Supplemental Table 1.
The conclusion that all three proposed pools should be used to assess PBMC fitness is not practical. Authors need to consider time and efforts in a clinical setting.
Answer 25. We revised the statement to “might”. (However, ELISPOT lends itself to high throughput testing, in particular in specialized laboratories. Our lab frequently tests hundreds of peptides/antigens individually on a single subject’s PBMC. One of our latest efforts along these lines included testing 553 individual peptides, plus 21 controls on individual subjects (please see DOI: 10.3390/v7082828 and https://doi.org/10.1101/2020.11.06.371633).
Font size and types are varied throughout the manuscript, and should be corrected.
Answer 26. We have unified it.
We apologize to Reviewer 3 for entering into in-depth discussions about how reliable epitope predictability is. Our different views on the interpretation of the data, should, however not detract from the data themselves. Of note, none of the two other reviewers too issue with our interpretation of the data and our skeptical view on the predictability of epitope utilization.
Round 2
Reviewer 1 Report
The authors have adequately addressed the raised comments.
Reviewer 3 Report
Authors have greatly enhanced the value of the manuscript by revising it. I agree with the authors’ response that manuscript has to be limited in scope (regarding allelic frequency covered). It is something authors may like to include in their discussion section. Also, I appreciate providing an in-depth analysis and perspective of the presented data. Supplementary figures and tables are helpful.
- It is a good idea to compare and provide details of what are the additional peptides (regions of CMV, Flu and EBV covered) in other peptide pools, than in CEF. Also, it is very logical that by enhancing numbers of peptides one has a better chance to identify responder T cells. However, it is worth discussing what is enough, or minimum numbers of peptides. A discussion on indications when and where what peptide pool should be applied, will be helpful. It should be noted that such diagnostic tools are being developed for clinical settings and cost is a big factor.
- For Fig. 2B, SFU could be presented as the standard “SFU/million PBMC”. Also, these are IFN-y secreting cells. So ELISPOT data could be presented as “# of IFN-y secreting cells/million PBMC” on Y-axis.